# Biologically Plausible Neural Networks via Evolutionary Dynamics and Dopaminergic Plasticity

## Abstract

Artificial neural networks (ANNs) lack in biological plausibility, chiefly because backpropagation requires a variant of plasticity (precise changes of the synaptic weights informed by neural events that occur downstream in the neural circuit) that is profoundly incompatible with the current understanding of the animal brain. Here we propose that backpropagation can happen in evolutionary time, instead of lifetime, in what we call neural net evolution (NNE). In NNE the weights of the links of the neural net are sparse linear functions of the animal's genes, where each gene has two alleles, 0 and 1. In each generation, a population is generated at random based on current allele frequencies, and it is tested in the learning task. The relative performance of the two alleles of each gene over the whole population is determined, and the allele frequencies are updated via the standard population genetics equations for the weak selection regime. We prove that, under assumptions, NNE succeeds in learning simple labeling functions with high probability, and with polynomially many generations and individuals per generation. We test the NNE concept, with only one hidden layer, on MNIST with encouraging results. Finally, we explore a further version of biologically plausible ANNs inspired by the recent discovery in animals of dopaminergic plasticity: the increase of the strength of a synapse that fired *if* dopamine was released soon after the firing.

## 1 Introduction

In his Turing award lecture, neural networks pioneer Geoff Hinton opined that *"evolution can't get gradients because a lot of what determines the relationship between the genotype and the phenotype is outside your control"* (Hinton, 2019). We beg to differ. Evolution does have what amounts to an effective oracle access to the (indeed, complex and intractable) mapping from genotype to phenotype. The well-established equations of population genetics governing evolution under recombination (Bürger, 2000; Chastain et al., 2014) describe the way whereby the distribution of genotypes in the population is updated from one generation to the next, informed by the empirical fitness of the phenotypes during lifetime; and these equations do bear a similarity to gradient descent and, even closer, to no-regret learning (Chastain et al., 2014). In this paper, we show that, in fact, quite effective training of neural nets can be carried out, without backpropagation, in evolutionary time.

The towering empirical success of ANNs has brought into focus their profound incongruity with what we know about the brain: backpropagation requires that and synaptic weights and plasticity be informed by downstream events. Clever versions of ANNs have been proposed recently that avoid this criticism: ANNs whose backward weights are random and fixed (Lillicrap et al., 2016) and a variant that also uses random feedback weights but with zero initial conditions (Nø kland, 2016), a backpropagation interpretation of STDP (a widely accepted theory of plasticity) (Bengio et al., 2015), unsupervised learning using STDP (Diehl & Cook, 2015), ANNs driven by neural competition (Krotov & Hopfield, 2019), or ANNs with target value propagation at each layer rather than the loss gradient (Lee et al., 2015).

Here we take a very different approach. We believe that, while forward neural computation is coterminous with life, backpropagation (i.e., feedback to individual neurons and synapses about their contribution to the overall performance of the circuit) can be effectively carried out over evolutionary

time. Suppose that the brain circuitry for a particular classification task, such as "food/not food", is encoded in the animal's genes, assuming each gene to have two alleles 0 and 1. A (haploid) genotype is a bit string. Crucially, we assume that the weight of each link of the neural network is a *fixed sparse linear function of the genes*. Evolution proceeds in generations. At each generation, a gene is an independent binary variable with fixed probability of 1. A population is sampled from this distribution of genotypes, and it experiences a sequence of inputs to the brain circuit. Fitness of each genotype depends, to some small degree, on the animal's success over its lifetime in the specific classification task. In the next generation, the allele frequencies will change slightly, depending on how each allele of each gene fared cumulatively (over both all inputs and all genotypes containing it) in the classification task. These changes follow the standard population genetics equations for the weak selection regime, see Bürger (2000); Chastain et al. (2014); weak selection means that the classification task is only one of the many biological functions (digestion, locomotion, other brain circuits, etc.) that affect the animal's fitness.

The question is, can competent brain circuits evolve this way? We offer theoretical evidence that this is indeed the case[1]. In Section 2 we prove that, if the classifier to be learned is linear, then evolution does indeed succeed in creating a neural network that classifies well, almost certainly and within polynomial (in the dimension of the problem) number of generations, individuals per generation, and neurons per individual. We also validate our theorem through experiments on the MNIST data set. Our experiments are not meant to compete with what is happening in the ANN world. We want to make the point that competent learning can happen in life: NNE with a single hidden layer already gives surprisingly good accuracy rates (more than $90\%$ accuracy for classifying the MNIST digits 0 to 4).

There is a different way of looking at, and motivating, our results, namely from the point of view of the study of the brain in connection to evolution. *"Nothing in biology makes sense except in the light of evolution,"* Theodosius Dobzhansky famously proclaimed. Neuroscientists have espoused this point of view, and evolutionary arguments come up often in the study of the brain, see for example Bosman & Aboitiz (2015). However, we are not aware of a technical discussion in the literature of the obvious existential question: Is the architecture of the brain susceptible to evolution through natural selection? Can brain circuits evolve? Our mathematical and empirical results in this paper on NNE strongly suggest that, indeed, effective brain circuits specializing in classification tasks *could have evolved.*

We also propose a second biologically plausible ANN-like mechanism, based on *dopaminergic plasticity*. It was recently established experimentally (Yagishita et al., 2014) that weights in certain synapses (in this case from the cortex to the striatum in the mouse brain, but not only) are increased if dopamine was released within 0.5-2 seconds after the synapse's firing. Intuitively, this is a reinforcement plasticity mechanism that "rewards" synapses whose firing led to a favorable outcome. Inspired by this experiment, we define dopaminergic neural nets (DNN), in which the weight of a link that fired (that is, both nodes fired during the current training example) is modified by a multiple of $(\frac{1}{4} - err^2)$, where $err$ is the error of the current training example. That is, links that fired are rewarded if the result was good, and actually also punished if it was not. Our experiments show that such DNNs can also learn to classify quite well, comparable to SGD.

**Our Contributions.** In Section 2, we give a rigorous proof that NNE with a single hidden layer succeeds in learning arbitrary linear target functions. In Section 3, we discuss experiments with NNE and DNN on MNIST.

## 2 ANALYSIS OF NNE

A genotype can be viewed as a vector $x \in \{0, 1\}^n$. A probability distribution over the genotypes is given by a vector $p \in [0, 1]^n$; a genotype $x$ is sampled by setting $x(i) = 1$ with probability $p(i)$, independently for each $i$. The neural network corresponding to a genotype $x$ is a *feed-forward neural network* (FFNN) whose weights are computed as follows. For a prediction network having $m$ links, the weights of the links are given by $Wx$, where $W$ is an $m \times n$ sparse *weight generation*

---

[1]We note incidentally that NNE is of course very much distinct from neuroevolution (see the recent survey Stanley et al. (2019)), which optimizes ANN architecture and hyperparameters through genetic algorithms.

*matrix*. We choose the entries of $W$ to be random and i.i.d.: with probability $\beta$, $W(i, j)$ is chosen uniformly at random from $[-1, 1]$, and is 0 with probability $1 - \beta$.

The input to the network is a vector $y$ drawn from a distribution $\mathcal{D}$ and has a label (possibly real-valued) $\ell(y)$. The output of the network on an input $y$ is $\mathsf{NNE}_x(y)$. In the simplest linear case (Section 2.1), $y \in \mathbb{R}^m$ and $\mathsf{NNE}_x(y) = x^T W^T y$. In our experiments (Section 3.1), we study the case when $\mathcal{D}$ is the uniform distribution over MNIST, and $\mathsf{NNE}_x(\cdot)$ is a 1-layer neural network with a ReLU output gate (see Section 3.1 for formal definition).

For each genotype $x$, we measure its performance by computing the loss $L(\mathsf{NNE}_x(y), l(y))$ (this could be squared loss, cross-entropy loss, etc.). For a probability distribution over genotypes $p$, we define the loss as

$$\mathcal{L}(p) := \mathsf{E}_{x \sim p} \mathsf{E}_{y \sim D} L(\mathsf{NNE}_x(y), \ell(y)).$$

We calculate the rewards $f^t(i)$ and $\bar{f}^t(i)$ as the expected negative loss whenever the allele is present and absent respectively.

$$f^t(i) = \mathsf{E}_{x \sim p^t} \left[ \mathsf{E}_{y \sim D} \left[ -L(\mathsf{NNE}_x(y), \ell(y)) \right] \mid x(i) = 1 \right]. \tag{1}$$

and

$$\bar{f}^t(i) = \mathsf{E}_{x \sim p^t} \left[ \mathsf{E}_{y \sim D} \left[ -L(\mathsf{NNE}_x(y), \ell(y)) \right] \mid x(i) = 0 \right]. \tag{2}$$

For the next generation we calculate,

$$p = p^t(i)(1 + \epsilon f^t(i)) \qquad \text{and} \qquad q = (1 - p^t(i))(1 + \epsilon \bar{f}^t(i))$$

We normalize $p$ and $q$ to make it a probability distribution. Thus the allele probabilities for the next generation will be,

$$p^{t+1}(i) = \frac{p}{p + q} = \frac{p^t(i)(1 + \epsilon f^t(i))}{1 + \epsilon \bar{f}^t(i) + \epsilon p^t(i)(f^t(i) - \bar{f}^t(i))}. \tag{3}$$

This is the standard update rule in population genetics under the weak selection assumption. The multiplier $\epsilon$ captures the small degree to which the performance of this task by the animal confers an evolutionary advantage leading to larger progeny.

Our first observation is that perfomance per allele is in fact a function of the gradient of the loss function.

**Lemma 1**

$$\mathcal{L}(p^t) = -\bar{f}^t(i) - p^t(i)(f^t(i) - \bar{f}^t(i)) \quad \text{and} \quad \frac{\partial}{\partial p^t(i)} \left( \mathcal{L}(p^t) \right) = -(f^t(i) - \bar{f}^t(i)).$$

*Proof.*

$$\begin{aligned}
\mathcal{L}(p^t) &= \mathsf{E}_{x \sim p} \mathsf{E}_{y \sim D} \left[ L(\mathsf{NNE}_x(y), \ell(y)) \right] \\
&= p^t(i) \mathsf{E}_{x \sim p^t} \left[ \mathsf{E}_{y \sim D} \left[ L(\mathsf{NNE}_x(y), \ell(y)) \right] \mid x(i) = 1 \right] \\
&+ (1 - p^t(i)) \mathsf{E}_{x \sim p^t} \left[ \mathsf{E}_{y \sim D} \left[ L(\mathsf{NNE}_x(y), \ell(y)) \right] \mid x(i) = 0 \right] \\
&= p^t(i)(-f^t(i)) + (1 - p^t(i)) \left( -\bar{f}^t(i) \right) = -\bar{f}^t(i) - p^t(i)(f^t(i) - \bar{f}^t(i)).
\end{aligned}$$

Here, the last line follows from equations 1 and 2. Now, taking the derivative w.r.t. $p^t(i)$ we get

$$\frac{\partial}{\partial p^t(i)} \left( \mathcal{L}(p^t) \right) = -(f^t(i) - \bar{f}^t(i)).$$

$\square$

We use this to prove the following theorem.

**Theorem 1** *Fix $\delta > 0$. Suppose $\nabla^2 \mathcal{L}(z) \preceq H \cdot I \ \forall z \in [0, 1]^n$. Let $U := \sup_{p \in [0,1]^n} \mathcal{L}(p)$ and $S_t := \{i \in [n] \mid \delta \leq p^t(i) \leq 1 - \delta\}$. For $\epsilon \leq \min\{1/(\max\{2U, 1\}), 2/H, 1\}$, there is an $\eta > 0$ s.t.*

$$\mathsf{E}(\mathcal{L}(p^{t+1})) \leq \mathcal{L}(p^t) - \eta \sum_{i \in S_t} \left( \nabla_i \mathcal{L}(p^t) \right)^2.$$

*Proof.* Using Equation 3 and Lemma 1 we get

$$p^{t+1}(i) - p^t(i) = \frac{\epsilon \cdot p^t(i)(1 - p^t(i))(f^t(i) - \bar{f}^t(i))}{1 + \epsilon \bar{f}^t(i) + \epsilon p^t(i)(f^t(i) - \bar{f}^t(i))} = -\frac{\epsilon \cdot p^t(i)(1 - p^t(i))}{1 - \epsilon \mathcal{L}(p^t)} \nabla_i \mathcal{L}(p^t)$$
$$= -\gamma_i \cdot \nabla_i \mathcal{L}(p^t)$$

where $\gamma$ is as defined above. For our choice of $\epsilon$, we have

$$1 - \epsilon \mathcal{L}(p^t) \geq 1 - \epsilon U \geq \frac{1}{2} \quad \text{and} \quad \epsilon \cdot p^t(i)(1 - p^t(i)) \leq \frac{\epsilon}{4} \leq \frac{1}{2H}.$$

Therefore, $\gamma_i \leq 1/H$. Using Taylor's theorem, there exists a $z \in [p^t, p^{t+1}]$ s.t.

$$\mathcal{L}(p^{t+1}) = \mathcal{L}(p^t) - (p^{t+1} - p^t)^T \nabla \mathcal{L}(p^t) + \frac{1}{2}(p^{t+1} - p^t)^T \nabla^2 \mathcal{L}(z)(p^{t+1} - p^t)$$
$$\leq \mathcal{L}(p^t) - \sum_i \gamma_i (\nabla_i \mathcal{L}(p^t))^2 + \frac{H}{2} \sum_i \gamma_i^2 (\nabla_i \mathcal{L}(p^t))^2 \qquad (\text{Using } \nabla^2 \mathcal{L}(z) \preceq HI)$$
$$= \mathcal{L}(p^t) - \sum_i \left( \gamma_i - \frac{H}{2}\gamma_i^2 \right)(\nabla_i \mathcal{L}(p^t))^2.$$

Since, $\gamma_i \leq 1/H$, we have $\gamma_i - \frac{H}{2}\gamma_i^2 \geq \gamma_i/2$. Therefore,

$$\mathcal{L}(p^{t+1}) \leq \mathcal{L}(p^t) - \sum_i \frac{\gamma_i}{2}(\nabla_i \mathcal{L}(p^t))^2 = \mathcal{L}(p^t) - \frac{\epsilon}{2(1 - \epsilon \mathcal{L}(p^t))} \sum_i p^t(i)(1 - p^t(i))\left(\nabla_i \mathcal{L}(p^t)\right)^2$$
$$\leq \mathcal{L}(p^t) - \frac{\epsilon \delta(1 - \delta)}{2(1 - \epsilon B)} \sum_{i \in S_t} \left(\nabla_i \mathcal{L}(p^t)\right)^2,$$

where $B := \inf_{p \in [0,1]^n} \mathcal{L}(p)$. Therefore, the conclusion of the theorem holds for $\eta = \epsilon \delta(1 - \delta)/(2(1 - \epsilon B))$. □

## 2.1 LEARNING LINEAR FUNCTIONS

In this section, we show that in the case of a linear target functions, with high probability, NNE converges to an allele distribution $p$ which is arbitrarily close to the correct linear labeling. Our NNE has $m$ input gates connected to one output gate (i.e., no hidden layers). For a genotype $x$, the weights of the connections are given by $Wx$. On input $y$, the NNE outputs $x^T W^T y$.

**Theorem 2** *Let $D$ be the uniform distribution over vectors in an $n$-dimensional unit ball. Let $a$ be a fixed vector with $\|a\| \leq 1$, such that the label of $y$ is $\ell(y) := a^T y$. Let $W$ have i.i.d. entries with $W_{ij} = \pm\sqrt{m/d}$ with probability $d/m$ and $0$ with probability $1 - (d/m)$. Then, for any $\delta \in (0, 1]$, with $n = O(m + \log(1/\delta))$, with probability at least $1 - \delta$, there exists an allele distribution $p$ s.t. $Wp = a$. Moreover, with probability at least $3/4$, for any $\epsilon \in (0, 1]$, with $n = \Omega(m(\log(1/\epsilon)/\epsilon^2)$, there is an $x \in \{0, 1\}^n$ s.t. $\frac{(Wx) \cdot a}{\|Wx\|\|a\|} \geq 1 - \epsilon$.*

We remark that the above guarantee works for *every* linear target function in $\mathbb{R}^m$. To learn, with high confidence, the target function from among $d$ unknown (arbitrary) linear functions, $m$ above can be replaced by $\log d$.

*Proof.* To have $Wp = a$, with $p(i) \in [0, 1]$, it suffices that the vector $a$ lies in the convex hull of the columns of $W$. This follows if the unit ball around the origin is contained in the convex hull of the vectors $W^{(i)}$. By duality, it suffices that every halfspace tangent to the unit ball (and not containing it) has at least one of the $W^{(i)}$ in it. For any single halfspace tangent to the unit ball, the probability that a random $W^{(i)}$ falls in it is at least a constant factor — each $W^{(i)}$ has squared length $m$ in expectation and concentrated near it. Thus, the halfspace it defines carves out a cap of constant measure. Next, by the VC theorem, if $n = \Omega(m + \log(1/\delta))$, with probability $1 - \delta$ every such halfpsace will contain a column of $W$. This establishes $\mathsf{E}(Wx) = a$.

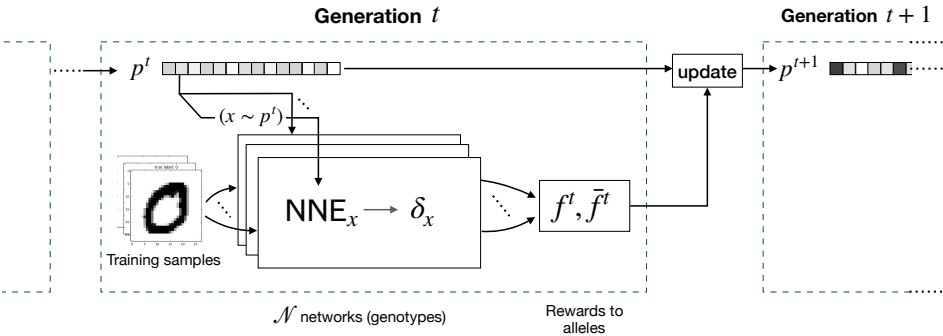

Figure 1: Trainng NNE across generations

To bound the error, we consider the subset of columns of $W$ that have a nontrivial inner product with $a$ and take their sum, i.e., let $J = \{i : w^i \cdot a \geq \frac{1}{\sqrt{m}} \|a\| \|w^i\|\}$ and $d = |J|$, and consider the random variables:

$$Y = \frac{1}{d} \sum_{i \in J} w^i \quad \text{and} \quad Z = Y \cdot a.$$

Then by the symmetry of the distribution of $W$, $\mathsf{E}(Y)$ points in the same direction as $a$ (in all other directions, the truncated distribution remains symmetric and therefore has mean zero). For convenience . Then,

$$\mathsf{Var}(Z) = \frac{1}{d} \sum_{i \in J} \mathsf{Var}(w^i \cdot a) \leq c\|a\|.$$

On the other hand,

$$\mathsf{E}(\|Y\|^2) = \frac{1}{d^2} \sum_{i,j \in J} \mathsf{E}(w^i \cdot w^j) \leq \frac{1}{d^2} \left( d\mathsf{E}(\|w^i\|^2) + d^2 \mathsf{E}(w^i \cdot a)^2 \right) \leq \frac{c_1 m}{d} + c_2 \|a\|.$$

where $c, c_1, c_2$ are absolute constants. So if $d = \Omega(m/\epsilon)$, then with large probability,

$$\frac{Y \cdot a}{\|Y\| \|a\|} \geq 1 - O(\epsilon). \tag{4}$$

However, we need this for every possible $a$. So we take an $(\epsilon/2)$-net of the unit ball in $\mathbb{R}^m$ (which has size at most $(3/\epsilon)^m$). For any fixed $a$, by taking $d = \Omega\left(\frac{m \log(1/\epsilon)}{\epsilon^2}\right)$, the Hoeffding bound tells us that the probability that (4) is violated is at most $e^{-\epsilon^2 d/4} \leq (\epsilon/4)^m$. Then, by a union bound, this bound on $d$ suffices for all $a$. Finally, with $n = \Omega(m \log(1/\epsilon)/\epsilon^2)$ whp, every cap $\{y : a \cdot y \geq 1/\sqrt{m}\}$ has at least $d$ columns of $W$ in it. $\qquad \square$

## 3 EXPERIMENTS

### 3.1 NNE ON MNIST

We study the effectiveness of NNE by evaluating its classification performance on the MNIST dataset.

To train an NNE via evolution of $T$ generations of genotypes, we fix a sufficiently large population size $\mathcal{N}$. Each generation $t \in [T]$ consists of a sample of $\mathcal{N}$ independently sampled genotypes from the allele distribution $p^t$, we denote this sample by $\mathcal{P}^t$. This distribution is updated based on the average performance $\bar{f}^t(i)$ and $\bar{f}^t(i)$ of all the genotypes on a task, in our case, MNIST handwirtten digit recognition task. We let the allele distribution $p^t$ evolve over $T$ generations in this manner (see fig. 1).

**Experimental setup.** We use 200 training samples for each of the digits, drawn uniformly at random from MNIST; we denote this set of training examples by $S$. $p^1$, the allele distribution for the first generation, is sampled uniformly at random from $[0, 1]^n$. We evaluate the performance of the alleles over $\mathcal{N} = 1000$ genotypes.

Our network has 784 input units, one hidden layer of $|h^1| = 1000$ units with ReLU activation and an output layer of 10 units with softmax activation. We add a sparse random graph between the input layer and the hidden layer: between a neuron in the input layer and a neuron in the hidden layer, we independently add an edge with probability $0.1$. The hidden layer is fully connected to the output layer. We choose $\beta = 0.0025$ for our experiments, i.e., each edge weight is a sparse random function of only $\beta$ fraction of the alleles. For the input sample $y$, $\ell(y)$ is now a one-hot encoding of the label, and $\mathsf{NNE}_x(y)$ is the soft-max output of the network. We use the cross-entropy loss function, $L(\mathsf{NNE}_x(y), \ell(y)) = -\sum_{c \in [C]} \ell(y)_c \log(\mathsf{NNE}_x(y)_c)$.

If a classifier were to randomly guess the label of an input intance, its loss function value would be $\alpha := -\log(1/10)$. We use the relative performance of the genotype w.r.t. to a random guess for our updates. To this end, we define for a genotype $x$, $\delta_x := \frac{1}{|S|} \sum_{s \in [S]} \max\{0, \alpha^2 - L(\mathsf{NNE}_x(y), \ell(y))^2\}$. For each allele, we calculate the rewards $f^t(i)$ and $\bar{f}^t(i)$ whenever the allele is present and absent respectively.

$$f^t(i) = \frac{\sum_{x \in \mathcal{P}^t} \delta_x x(i)}{\sum_{x \in \mathcal{P}^t} x(i)} \qquad \text{and} \qquad \bar{f}^t(i) = \frac{\sum_{x \in \mathcal{P}^t} \delta_x (1 - x(i))}{\sum_{x \in \mathcal{P}^t} (1 - x(i))}.$$

The allele distribution for the next generation is updated using equation 3.

NNE as described above achieves **78.8%** test accuracy on the full MNIST test set. While this is somewhat far from the state of art in classification of MNIST images, our results demonstrate that very basic NNEs can perform reasonably well in this task.

**Convergence of allele distributions.** We repeat for many (hundreds of thousands) generations. As our theoretical results predict (see also Mehta et al. (2015)), the vast majority of genes have allele probabilities that are very close to $0$ and $1$. Figure 2 shows the fraction of allele probabilities that are at a distance $[x, 1-x]$ from $0$ or $1$, i.e., $y$ is calculated as $y = 1 - \frac{|\{i : \min\{p^t(i), 1 - p^t(i)\} \le x\}|}{n}$.

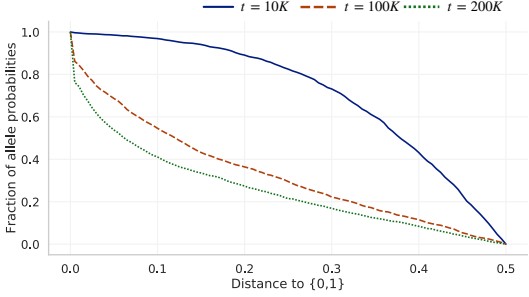

Figure 2: Convergence of allele distribution after $t$ generations: x-axis shows the distance of the allele probabilities from $0$ or $1$ and the y-axis shows the fraction of $n$ allele probabilities that are at a distance $[x, 1-x]$ from $0$ or $1$.

**NNE with output layer training.** The biological implausibility objection of using stochastic gradient based updates is less acute for the output layer, since in animal brains synaptic changes due to plasticity happen at the post-synaptic neuron, and for the output layer this is the output neuron. Even then, computing exact (or approximate) gradients is a nontrivial computational task; instead we consider using just the *sign* of the gradient for only the output layer as a lifetime training mechanism.

For the same network described as above, we randomly initialize the network weights using allele distribution learned using the NNE. We then calculate the sign of the gradient of the output layer weights and update the weights in the opposite direction (SignSGD), using a sufficiently small learning rate $\epsilon'$, similar to stochastic gradient descent. For $i$ in the hidden layer and $j$ in the output

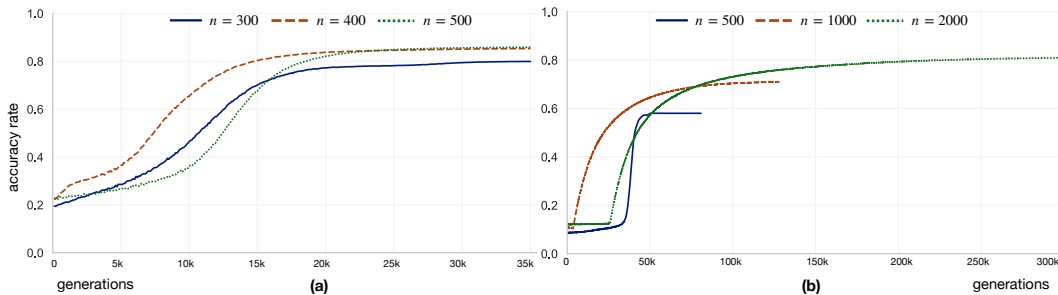

Figure 3: Number of genes ($n$) vs performance of NNE: **(a)** Accuracy rates of NNE on MNIST $0-4$ showing the effect of number of genes on performance. **(b)** Similarly, we also plot the accuracy rates of NNE on MNIST $0-9$ dataset while varying the number of genes. The accuracy trends show that more the number of genes, better the performance of NNE, but at the cost of more training time.

| model | MNIST 0 to 4 | MNIST 0 to 9 |
|---|---|---|
| NNE | 92.1 | 78.8 |
| NNE + SignSGD | 91.6 | 85.6 |
| SGD | $96 \pm 0.4$ | $88.2 \pm 0.75$ |

Table 1: Accuracy rates of NNE on MNIST test digits.

layer, the update is

$$w_{ij} := w_{ij} - \epsilon' \cdot sign\left((z_j - \ell(y)_j)h_i\right) \tag{5}$$

where $h_i$ is output of the neuron $i$, and $z_j$ softmax output of neuron $j$ (see appendix for the proof). SignSGD has been shown to be effective for traning large deep neural networks (for e.g., see Bernstein et al. (2018)).

We perform a few hundred iterations of this training using batch size 50. In this experiment (NNE + SignSGD), we obtain **86.3**% accuracy on full MNIST test set. This further demonstrates that biologically plausible neural networks can perform reasonably well in this task.

**Number of genes.** A crucial choice for an NNE is the number of genes. In our experiments, we use a few thousand genes; this is not unreasonable as it is estimated that about $5,000$ genes are expressed in the cells of the mammalian brain. To investigate further, we compare the performance of our algorithm with increasing values of $n$ (the number of genes). Figure 3 presents the validation accuracy trends on the same network described above for five class $[0-4]$ classification and for full MNIST dataset. We observe that the accuracy rate of the network improves significantly with increase in the number of genes. However, it requires much longer training time to achieve a desired accuracy rate.

Table 1 compares the results of all the models along with the baseline, stochastic gradient descent trained on the same subset of MNIST.

## 3.2 Dopaminergic Neural Nets (DNNs)

DNNs are biologically plausible ANNs based on dopaminergic plasticity. They learn by a weak form of immediate reinforcement - "rewarding" synapses whose firing led to a favourable outcome. If a connection between two neurons has fired during a training step, then its weight is increased if the square error was low (less than $\frac{1}{4}$). In this section, we demonstrate that simple DNNs can perform reasonably well for tasks like classifying the images in the MNIST dataset.

**Experimental setup.** For our experiments we use a network consisting of an input layer, a single hidden layer, and an output layer consisting of $784$, $h$, and 10 neurons respectively. Each neuron in the input layer has a link to each neuron in the hidden layer, and its weight is initialised by the popularly used Kaiming Uniform (more commonly called He initialisation (He et al., 2015)). These weights are unchanged through out the learning process. Recent theoretical results suggest that a

| $h$ | SGD | SignSGD | DNN |
|---|---|---|---|
| 1000 | 90.34 | 86.84 | 84.84 |
| 100,000 | 92.56 | 90.21 | 90.76 |

Table 2: Test accuracies of different models for different $h$.

large enough random layer is sufficiently rich and efficiently trainable (Vempala & Wilmes, 2019) (see also (Rahimi & Recht, 2008)).

Each neuron in the hidden layer has a link to each neuron in the output layer. The output layer outputs the softmax score. The weights of this layer are learned using plasticity based updates. On seeing an input $y$, the DNN tries to predict the label of $y$; let us denote this by $\mathsf{DNN}_W(y)$. If the DNN got the prediction correct, i.e. the loss $L(\mathsf{DNN}_W(y), \ell(y))$ is at most $\epsilon_0$, then weight $w_{ij}$ get increased by a small amount, provided the output neuron $j$ has low error (i.e. $|z_j - l(y)_j|^2 \leq 1/4$) where $z_j$ is the $jth$ coordinate of $\mathsf{DNN}_W(s)$.

Formally, the update rule is as follows for $i$ in the hidden layer and $j$ in the output layer.

$$w_{ij} = w_{ij} + \epsilon_1 \frac{\max\left\{0, \frac{1}{4} - |z_j - \ell(y)_j|^2\right\} \cdot \max\left\{0, L(\mathsf{DNN}_W(y), \ell(y)) - \epsilon_0\right\}}{\left(\frac{1}{4} - |z_j - \ell(y)_j|^2\right) \cdot (L(\mathsf{DNN}_W(y), \ell(y)) - \epsilon_0)}.$$

**Experimental results.** To study the effectiveness of our DNN in the 10-class MNIST digit classification, we compare its peformance with some other standard baselines.

1. SGD: In this we use the standard stochastic gradient descent (with the Adam optimiser Kingma & Ba (2015)) based updates to train our network.
2. SignSGD: As before, we use the sign of the gradient for updates (equation 5).

Table 2 shows the results for different $h$ values. All results are after 500 epochs of training. As with NNE we use the cross-entropy loss for all the models. We found that $\epsilon_0 = 0.75$ and $\epsilon_1 = 1$ for the DNN gives reasonable performance. Our DNN gives encouraging results and is comparable to SignSGD in performance.

## 4 DISCUSSION AND FURTHER WORK

We have presented two biologically plausible mechanisms for the evolution of neural networks, motivated by the brain, and based on evolution. One feature of these mechanisms is that they process one input instance at a time (i.e., unit batch size) and do not require the computation of gradients.

Our preliminary experiments with bio-plausible mechanisms suggest that they are promising alternatives to backpropagation and the explicit use of gradients. The results raise several interesting possibilities:

1. In our current set-up, network weights are sparse *linear* functions of alleles. What if we used nonlinear functions (e.g., sigmoids or ReLUs) to define the weights?
2. In our experiments, allele distributions approach $0/1$ values for most coordinates. We could take advantage of this by fixing alleles that are sufficiently close to 0 or 1 and continue only on the rest.
3. Can this approach be used, with greater success, for multi-layer networks?
4. Does NNE or DNN implicitly optimize the underlying *architecture*?
5. A recent model of memory creation and association in the mammalian brain is based on plasticity and *inhibition* (Papadimitriou & Vempala, 2019). In this model, inhibition is implemented as a cap, where the top $k$ highest weighted input neurons of an entire layer are the ones that fire; the rest of the neurons in the layer are suppressed. Our experiments with DNN indicate that using a $k$-cap for the hidden layer with $k$ as small as 256 when $h = 100,000$ does not degrade performance (and even enhances it slightly), while reducing computation. Can $k$-cap help with learning?

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

## 5 APPENDIX

**Lemma 2** *For neuron $i$ in the hidden layer and neuron $j$ in the output layer, the gradient of the cross entropy loss with respect to the weights $w_{ij}$ in the final layer is*

$$\frac{\partial L}{\partial w_{ij}} = (z_j - \ell(y)_j)h_i.$$

*where $h_i$ is output of the neuron $i$, and $z_j$ softmax output of neuron $k$.*

***Proof.*** For the hidden layer output $h$ and the final layer weights $W$, we compute the soft-max output as follows,

$$o = W^T h.$$
$$z = softmax(o).$$

True output $l(y)$ is a one-hot encoded vector and the output of the model $z$ is the *softmax* of the final layer input $o$. The derivative of loss function $L$ w.r.t. $o$ is given by

$$\frac{\partial L}{\partial o_j} = -\sum_{k=1}^{C} \frac{\partial \ell(y)_k \log(z_k)}{\partial o_j}$$

$$= -\sum_{k=1}^{C} \ell(y)_k \frac{\partial \log(z_k)}{\partial o_j} = -\sum_{k=1}^{C} \ell(y)_k \frac{1}{z_k} \frac{\partial z_k}{\partial o_j}$$

$$= -\frac{\ell(y)_j}{z_j} \frac{\partial z_j}{\partial o_j} - \sum_{k \neq j}^{C} \frac{\ell(y)_k}{z_k} \frac{\partial z_k}{\partial o_j} = -\frac{\ell(y)_j}{z_j} z_j(1 - z_j) - \sum_{k \neq j}^{C} \frac{\ell(y)_k}{z_k}(-z_k z_j)$$

$$= -\ell(y)_j + \ell(y)_j z_j + \sum_{k \neq j}^{C} \ell(y)_k z_j = -\ell(y)_j + \sum_{k=1}^{C} \ell(y)_k z_j = -\ell(y)_j + z_j \sum_{k=1}^{C} \ell(y)_k = z_j - \ell(y)_j.$$

We know that $o_j = \sum_i w_{ij} h_i$. Hence,

$$\frac{\partial o_j}{\partial w_{ij}} = h_i.$$

Then the gradient for the final layer weights $w_{ik}$ is calculated as:

$$\frac{\partial L}{\partial w_{ij}} = \frac{\partial L}{\partial o_j} \cdot \frac{\partial o_j}{\partial w_{ij}} = (z_j - \ell(y)_j)h_i.$$

$\square$

