# OpenReview forum: "Biologically Plausible Neural Networks via Evolutionary Dynamics and Dopaminergic Plasticity"
_ICLR.cc/2020/Conference — Reject_

### Official Review · AnonReviewer2 · 2019-10-11
**Official Blind Review #2**

**Rating:** 1

**Review:**

First of all, I must confess that my knowledge is quite limited to read this paper. Perhap the authors present something that I can not catch up at the present.

I conjecture the paper would like to bring the evolution in genetics and perhap brain cirecuits as well to define a novel neural net model, called NNE by the authors.
The paper is somewhat cumbersome in the introduction that makes the reader (here is myself) can not understand the main idea. At first the authors introduce about evolution in genetics and genomics that is a bit different from what I known. Then the authors claim that they can show that the brain circuits can evolved in their model.
There are so many mistake and/or typos in sentences and in mathematical formulation. These make me can not finish reading the paper. I have to stop reading at the end of Section 2.
Here are my concerns and questions:
1). What is "STDP" in the 2nd papragraph in Introduction?
2). in the 3rd papragraph in Introduction: "Suppose that the brain circuitry for a particular classiﬁcation task, such as “food/not food”,is encoded in the animal’s genes, assuming each gene to have two alleles 0 and 1". This is realistics, the allels in animal is 0 1 or 2 if encoded.
3). The authors use the words "gene, genotype, phenotype" in a special way that is different to what I known in genomics (in GWAS).
4).  in the 3rd papragraph in Introduction: "At each generation, a gene is an independent binary variable with ﬁxed probability of 1". What do you mean by fixed probability of 1? I can not understand in anysense that I know.
5). In Section 2, What is n? the authors start the mathematical formulation but I can not find out what is n? Is it the sample size?
6). In Section 2, paragraph 2, you define y~ \mathcal{D}, BUT then in all formulas later you denote y ~ D. What is D??? I can not understand.
7).  In Section 2, paragraph 2, you define a label of y as \ell(y), BUT then in the 1st sentence of the 3rd paragraph in Section 2 you wrote    L(NNE_x(t), l(y)). What is l(.) here ???
8). In the equations (1) and (2), what is p^t  ???  You have NOT defined it.
9). Right after equations (1) and (2), What is \epsilon ???? Can NOt understand.
10). The sentence right after the equation (3): "This is the standard update rule in population genetics under the weak selection assumption." This is NOT trivial to me, and even the machine learning comunity, we do not know this rule, it is not obvious. PLEASE provide exact reference.
11). the first equation in the PROOF of LEMMA 1 wrong  \mathcal{L} (p^t)  should be equal to E_{x~p^t}  NOT p.
12). in the PROOF of Theorem 1, I can NOT find out where \gamma has been defined?
13). What is d  in Theorem 2? is it the dimension? I make too many guesses !!!


**Experience Assessment:**

I have read many papers in this area.

**Review Assessment: Checking Correctness Of Derivations And Theory:**

I carefully checked the derivations and theory.

**Review Assessment: Checking Correctness Of Experiments:**

I did not assess the experiments.

**Review Assessment: Thoroughness In Paper Reading:**

I read the paper at least twice and used my best judgement in assessing the paper.

---

> ### Author Response · Authors · 2019-11-14
> **Response to Reviewer #2**
>
> We thank you for your valuable comments. We provide the answers below for the concerns raised in the review:
>
> In this paper, we propose two approaches to understand the process of learning in animal brain, that opens up the exciting possibility of improved learning using insights from evolution and the brain; 1) NNE (Neural Net Evolution), inspired by the standard update rule in population genetics, that succeeds in creating neural networks that perform simple labelling tasks with modest but promising preliminary results, both theoretical and experimental, suggesting that the neural networks in animal brain could have evolved, 2) DNN (Dopaminergic Neural Network) that is inspired from the recently established results on dopaminergic plasticity, which also give promising results on the classification tasks, supporting the plausibility of plasticity based updates in animal brains.
>
> The approaches used in this paper are consistent with evolution in genetics, and we provide explanations and corrections to the minor issues raised in the review. Nevertheless, these issues don’t seem significant enough to hinder the understanding of the methods proposed.
>
> (1)
> STDP is an acronym for Spike-Time Dependent Plasticity. This appears in all the relevant works cited in the introduction section of the paper.
>
> (2)
> We consider a simple case when the alleles are 0 and 1. In case of more than 2 possible alleles, our method can be easily extended by appropriately encoding the alleles.
>
> (3)
> We define gene as a single bit of information with two alleles 0 and 1 (section 1, paragraph 3), and genotype as a string of alleles (section 2, paragraph 1); both these definitions are consistent with those of genomics. We don’t use the term phenotype explicitly in our method.
>
> (4)
> As mentioned in section 1, paragraph 3, each gene has two alleles 1 or 0. At each generation, we fix the allele probabilities. Hence, the probability of a gene i having allele 1 for the genotypes of that generation is fixed. This sentence shall be updated in the revised version of the paper to remove the ambiguity.
>
> (5)
> As mentioned in section 2, paragraph 1, n is the number of genes for each genotype. Hence, each genotype is a binary vector of size n, i.e, x \in {0,1}^n.
>
> (6)
> This is a minor typo. We correct it to y ~ D everywhere for consistency.
>
> (7)
> This is a minor typo. We correct it to L(NNE_x(t), \ell(y)).
>
> (8)
> p^t is the allele probability distribution at generation t. We will add some explanation to make this clear.
>
> (9)
> As mentioned in the sentence after equation (3), “The multiplier \epsilon captures the small degree to which the performance of this task by the animal confers an evolutionary advantage leading to larger progeny.” \epsilon can be viewed as the “learning rate”.
>
> (10)
> We have provided exact reference to this sentence multiple times in the introduction - [Burger (2000); Chastain et al. (2014)] in section 1 paragraph 1 and 3. We also briefly explain in section 1 paragraph 3 what “weak selection” means.
>
> (11)
> This is a minor typo.
>
> (12)
> \gamma is implicitly defined and can be calculated from the last equality in the first equation array on page 4.
>
> (13)
> d is defined in the proof of theorem 2 to be the size of the set J, which is again defined in the proof of theorem 2.

---

### Official Review · AnonReviewer1 · 2019-10-22
**Official Blind Review #1**

**Rating:** 3

**Review:**

In this paper the authors propose a method for training neural networks using evolutionary methods. The aim of developing this method is to provide a biological alternative to back-propagation. The authors prove that their method converges and with high probability succeeds in learning linear classification problems. Another method is also proposed which is linked to dopaminergic neurons.

In terms of presentation, the paper is generally clear and well-written. I was not able to assess the importance of the theoretical contributions of the work as my research is not in this area, so my comments are limited to the other aspects.

With regard to the biological plausibility of the method, it is unclear to me how the evolutionary method proposed here can enable learning in typical scenarios such as conditioning experiments in animals. The learning processes in animals typically occurs in short time spans (for example a few training sessions for conditioning to stimuli predicting food/no food) and therefore I don’t find it plausible to suggest evolutionary methods across generations are behind such forms of learning. Perhaps what the authors have in mind applies more to other forms of behaviour such as innate and involuntary responses in animals formed across generations rather than ongoing updates in synaptic plasticity as an animal adjusts its behaviour using environmental feedback. But then in this case the biological plausibility of the method seems fairly limited and not really an alternative to methods such as back-propagation.

The other biological aspect of the proposed work is the connection to dopamine and using the sign of gradients for updating the weights. I think connecting the current learning rule to the activity of dopamine neurons requires quantitative comparisons with experimental data, otherwise although I agree that the method is biologically inspired, but whether it is biologically plausible is not clear.

Based on the above comments, I think the work will benefit from further developments before being ready for publication.

**Experience Assessment:**

I do not know much about this area.

**Review Assessment: Checking Correctness Of Derivations And Theory:**

I did not assess the derivations or theory.

**Review Assessment: Checking Correctness Of Experiments:**

I assessed the sensibility of the experiments.

**Review Assessment: Thoroughness In Paper Reading:**

I read the paper at least twice and used my best judgement in assessing the paper.

---

> ### Author Response · Authors · 2019-11-14
> **Response to Reviewer #1**
>
> We thank you for your valuable comments. We provide the answers below for the concerns raised in the review.
>
> (1)
> In section 2 we prove that, if the classifier to be learned is linear, then evolution indeed succeeds in creating neural network that classifies well, i.e., we show that the necessary updates to synapses, to perform the classification task, could happen in light of evolution. Our experimental results in table 1 on MNIST support this. We are not suggesting that the evolutionary mechanism is employed during a lifetime, only part of it is, to elicit feedback. Our paper takes the approach that evolutionary dynamics inspires a new type of ANN.
>
> (2)
> As mentioned in page 2 paragraph 4 in our submission, one of the recent investigations [Yagishita et al., 2014)] reveals that the release of dopamine affects the structural plasticity of certain synapses, within a narrow period of time after the synapse’s firing. Since the proposed DNN implements a very similar mechanism – the release of dopamine is captured by the favorable outcome (via error) and the synaptic update is a function of this favorable outcome – it could be considered biologically plausible.

---

### Official Review · AnonReviewer3 · 2019-10-24
**Official Blind Review #3**

**Rating:** 3

**Review:**

This paper argues that Artificial Neural Network (ANN) lack in biological plausibility because of the back-propagation process. Therefore, the authors provide an alternative approach, named neural net evolution (NNE) that follows evolutionary theory. This approach uses a large number of genotypes (in the form of vector with binary logits) that will evolve overtime during training. It does not require to calculate the gradient explicitly. The authors have conducted some experiments on MNIST using ANN with only one hidden layer. The experimental results show that the NNE can learn the classification task reasonably well considering that no explicit back propagation is used.

I think overall the motivation to combine ANN with evolutionary theory is very interesting. The reviewer is not very familiar with evolutionary theory. So I judge this paper in the perspective of machine learning, from which I think the current approach is a week variant of back-propagation that still relies on gradient (see detailed comments below). Based on this, I give my rating.

The approach is formulated as NNE_x(y) = (x^T)*(W^T)*y. In traditional linear regression, W is the weight to be learnt. In this paper's formulation, W is named as a weight generation matrix, which is choosing to be random and i.i.d. with certain probabilities. The parameters to be optimized is x, which is named as a genotype that is viewed as a vector x \in {0, 1}^n. So first of all, as W is fixed so the formulation is very similar to a traditional linear regression with an additional linear transform. The difference is that x is a binary vector with probabilities. These probabilities are optimized over time.

From the Equations 1), 2) and 3), the probabilities are updated in a way to minimize the loss. This is kind of similar to back-propagation. Then the probabilities are updated and thus x is changed as well. In my understanding, this is still gradient-based optimization. I do not see it fundamental different to back-propagation. This is my main concern about this work.

I did not check the details of Theorem 1. Could the authors please comment what is the purpose of Theorem 1 before proving it? This part is unclear to me in this paper.

One more question, for the W matrix,  the authors choice beta = 0.0025 in the experiment. Is there any particular reason for this choice? Or does it matter what value to choice as it is fixed anyway?


**Experience Assessment:**

I do not know much about this area.

**Review Assessment: Checking Correctness Of Derivations And Theory:**

I assessed the sensibility of the derivations and theory.

**Review Assessment: Checking Correctness Of Experiments:**

I assessed the sensibility of the experiments.

**Review Assessment: Thoroughness In Paper Reading:**

I read the paper at least twice and used my best judgement in assessing the paper.

---

> ### Author Response · Authors · 2019-11-14
> **Response to Reviewer #3**
>
> We thank you for your valuable comments. We provide the answers below for the concerns raised in the review.
>
> (1)
> The point of this paper is not to produce a new machine learning framework for classification tasks, but to understand how the animal brain could work, by studying biologically plausible neural networks under the lens of machine learning.
>
> (2)
> Our approach doesn’t always have to be used for a linear classifier NNE_x(y) = x^T W y, the NNE could be any function (see Lemma 1 and Theorem 1). We analyze the linear NNE in Theorem 2 to show that for any linear target function, NNE converges to an allele distribution arbitrarily close to the true labelling function, with high probability. The parameters that are optimized during the training process of NNE are the allele probabilities p. x is sampled from p as mentioned in section 2, paragraph 1. After sampling x, we use the weight generator matrix W of each layer to generate weights Wx of that layer of NNE. Updates to the allele probabilities p indirectly update the weights of all the layers of NNE simultaneously. Hence, although in linear case, NNE formulation looks similar to linear regression, in multilayer NNE, it’s not.
>
> (3)
> The way the probabilities are updated is given in equation 3, this is consistent with the weak selection regime as mentioned in section 1, paragraph 3 of our paper. While there is not explicit back-propagation, our point is that weak selection is implicitly performing something similar in spirit (this is not vanilla back-propagation since the weights of the links in the networks cannot be updated directly).
>
> (4)
> In lemma 1 we show that the performance of the allele, f, is in fact a function of the gradient of the loss function. In gradient descent-based optimization, it is well known that the decrease in the value of loss function after each iteration is proportional to the squared norm of the gradient (for e.g., see section 9.3 in [1]). Theorem 1 says that, given the update rule of the allele probabilities in equation (3), something similar holds for NNE! By choosing a small enough learning rate, the expected decrease in the loss at generation t+1 is proportional to the squared norm of the gradient at generation t taken at the coordinates of the allele distribution which are far from 0 or 1.
>
> (5)
> Beta defines the sparsity of each weight in the weight generation matrix. We want each weight w_ij of the network to be a sparse random combination of the genes x so that update to an allele probability p(i) affects only a small number of synapses in the network that depend on x(i).
>
> References:
>
> [1]  Stephen Boyd and Lieven Vandenberghe. 2004. Convex Optimization. Cambridge University Press, New York, NY, USA.

---

### Decision · Program_Chairs · 2019-12-19

**Decision:**

Reject

**Comment:**

Unfortunately the paper is confusingly written, and there is only agreement by all reviewers on the rejection of the paper.  Indeed, if all reviewers and the area chair do not interpret the paper well, the authors' best response would be to rewrite the papers rather than disagree with all reviewers.

In the area chair's opinion, the current form the paper does not merit publication.  The authors are advised to address the reviewers' concerns, rework the paper, and submit to a conference again.